# Comparison of CNN models on a multi-scanner database in colon cancer histology

**Petr Kuritcyn**[1]                                        PETR.KURITCYN@IIS.FRAUNHOFER.DE
**Michaela Benz**[1]                                       MICHAELA.BENZ@IIS.FRAUNHOFER.DE
**Jakob Dexl**[1]                                                    DEXLJB@IIS.FRAUNHOFER.DE
**Volker Bruns**[1]                                         VOLKER.BRUNS@IIS.FRAUNHOFER.DE
[1] *Fraunhofer Institute for Integrated Circuits IIS*

**Arndt Hartmann**[2]                                      ARNDT.HARTMANN@UK-ERLANGEN.DE
**Carol I. Geppert**[2]                                       CAROL.GEPPERT@UK-ERLANGEN.DE
[2] *Institute of Pathology, University Hospital Erlangen*

## Abstract

One of the most important challenges for computer-aided analysis in digital pathology is the development of robust deep neural networks, which can cope with variations in color and resolution of digitized whole-slide images (WSIs). It has been shown that color augmentation during training is a useful method to aid a model generalize better to heterogeneous data. In this work, we compare several state of the art models on a multi-scanner database comprising slides each digitized with six different scanners. All of the networks are trained with data of only one scanner applying a combination of color and blur augmentation techniques. All models show similar tendencies across the different scanner databases but differ in the overall classification accuracy. Differences in training and inference time, however, are more pronounced: on a mid-range GPU, the inference time of the fastest model (Quick-Net) is 13 times faster than the slowest one (EfficientNet B4). There is also a trade-off between speed and accuracy, the slower networks are more stable across different scanners and show the overall best performance. A good compromise between quality and inference time is achieved by EfficientNet B0.

**Keywords:** Histopathology; Data Augmentation; Tissue Classification; CNN

## 1. Introduction

Digitization of slides in computational pathology is a crucial step, which may introduce significant variations in color and resolution due to the use of different scanners. In our previous work (Kuritcyn et al., 2021) we addressed this challenge by applying different augmentation techniques during training and tested on our multi-scanner database of 30 slides each digitized with six different scanners. Another challenge in digital pathology is the huge size of whole-slide images, which can consist of several giga pixels and lead to high computation times. Therefore, in this work, we compare different state-of-the-art networks (Three versions of EfficientNet: B0, B3 and B4 (Tan and Le, 2019), Xception, an adapted version of Xception, Inception, ResNet, DenseNet, MobileNet and QuickNet) in terms of their robustness and inference time.

## 2. Materials and Methods

Our dataset consists of 161 hematoxylin and eosin (H&E) stained colon tissue sections with manual annotations of seven tissues classes (tumor cells, mucosa, etc.). The data for training and validation is derived from 122 WSIs, acquired with a 3DHISTECH MIDI scanner, resulting in 2,173,515 labeled image patches with a size of 224 x 224 pixel for training and 719,000 patches for validation. A disjoint set of 30 slides was scanned with six different scanners and annotations were automatically transfered resulting in scanner-specific test datasets each comprising more than 500,000 image patches. The resolution of the scanners varies from 0.17 to 0.35 µm/pixel and significant color variations are present. A more detailed description of the datasets is given in (Kuritcyn et al., 2021).

Training was carried out on a NVIDIA Tesla P100 using the TensorFlow framework. The batch size was set to 105 for all models except EfficientNet B3, B4 and QuickNet, where it was decreased to 35 due to GPU memory constraints. For all models, the Adam optimizer with a learning rate of 0.001 and an exponential decay was used. Each network was trained three times and test results were averaged (see Figure 1). We introduced color variance in the training data using a combination of hue, saturation and H&E color augmentations (Tellez et al., 2019). Additionally, we added a blur augmentation to counter the presence of out of focus regions in some WSIs. No geometric augmentations were applied. Besides the standard Xception model we also trained an adapted version (Xception adapt) described in (Kuritcyn et al., 2021). Inference tests were done with a mid-range NVIDIA GeForce GTX 1060 GPU with 6 GB memory using the TensorFlow 2.3 C API with a batch size of 30 and averaging over 5275 batches. Training and inference time are presented in Figure 2.

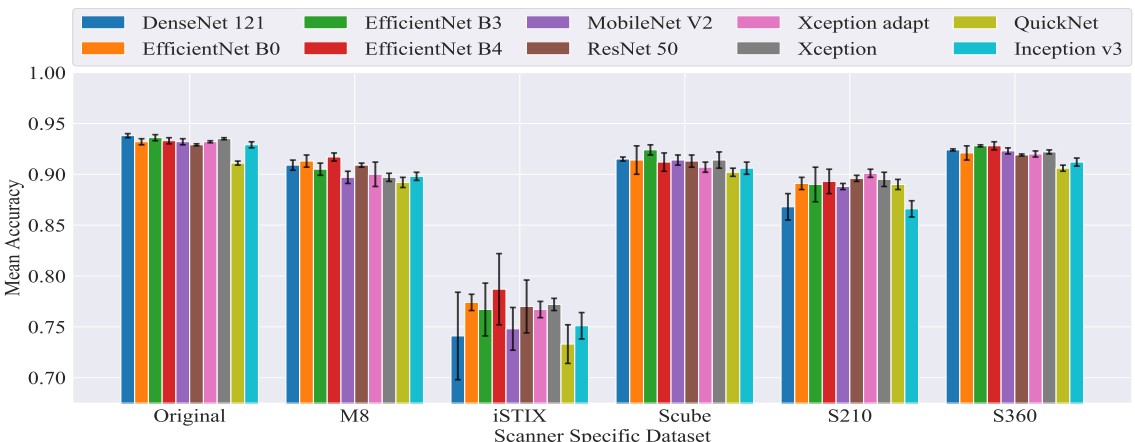

Figure 1: Average classification accuracy on the different scanner test datasets. All models were trained on the 3DHISTECH MIDI scanner (Original).

## 3. Results and Conclusion

Most models achieve recognition rates around 90% on all except the iSTIX dataset. Similarly, the standard deviation on all datasets is lower than for iSTIX. In comparison the latter has a poorer image quality due to the nature of the manual scanning process. Compared

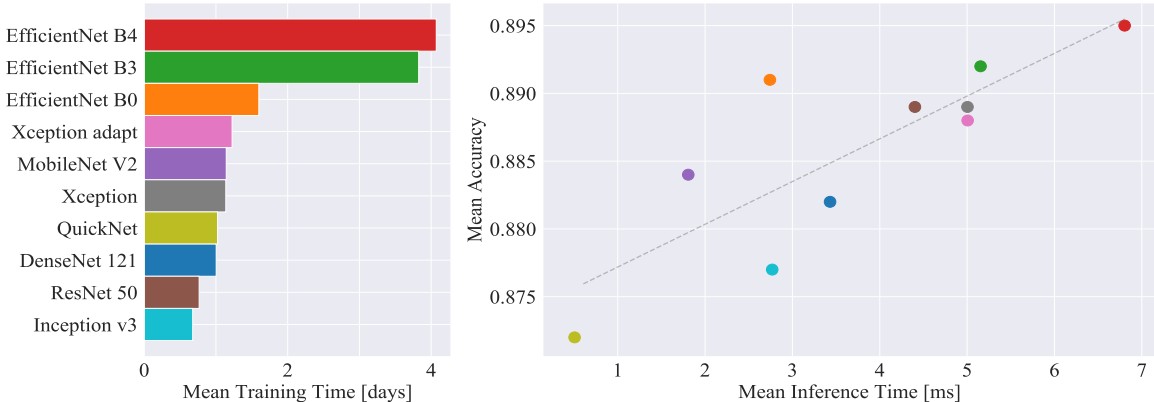

Figure 2: Mean training time until early stopping (left). Mean classification accuracy over all datasets plotted against the mean inference time per image patch (right).

to our previous work the additional blur augmentation significantly increases the accuracy on the iSTIX dataset from 0.621 to 0.761 (Xception adapt). The three EfficientNet models achieve the highest mean accuracies over all datasets. In (Tan and Le, 2019), however, e.g. Xception performs better on the ImageNet dataset than EfficientNet B0. This shows that ImageNet ranking is not directly transferable to the domain of histopathology. A decision on which model presents the best trade-off between accuracy and inference speed depends on how these attributes are weighted. In an attempt to find an objective score with equal weights for quality and speed, we propose to normalize both dimensions to a range of 0 (slowest/least accurate) to 1 (fastest/most accurate) and then average both values for each model. EfficientNet B0 shows the best overall score with 0.73, while all other models score between 0.42 (Inception v3) and 0.65 (MobileNet).

## Acknowledgments

This work was supported by the Bavarian Ministry of Economic Affairs, Regional Development and Energy through the Center for Analytics – Data – Applications (ADA-Center) within "BAYERN DIGITAL II" and by the BMBF (16FMD01K, 16FMD02 and 16FMD03).

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
