# OpenReview forum: "Comparison of CNN models on a multi-scanner database in colon cancer histology"
_MIDL.io/2021/Conference/Short — MIDL 2021 Poster_

### Official Review · Reviewer_CHrW · 2021-04-22

**Confidence:** 4
**Final Rating:** 3

**Summary:**

This paper compares different CNN architectures EfficientNet, Xception, Inception, ResNet, DenseNet, MobileNet, and QuickNet on a multi-scanner database comprising slides each digitized with six different scanners both in terms of classification accuracy and inference time.

The results show that EfficientNet B0 achieves the best trade-off between accuracy and inference time by achieving the best overall score of 0.73. Besides, it seems the images in the iSTIX dataset provide poorer image quality compared with other scanners.

**Strengths:**

* Comparing image quality of different scanners through classification accuracy of CNNs has practical value in the field of computational pathology

* Experiments are extensive

* The model that presents the best trade-off between accuracy and inference has been identified





**Weaknesses:**

* As a validation paper, this is necessary to conduct a hypothesis test on experiments and check if the differences are statistically significant. In the current paper, this is not clear for example the difference between two scanners or CNN architectures is significant.

* Some CNNs have been used but not cited. e.g.,   Xception, Inception, ResNet, DenseNet, MobileNet, and QuickNet

* multi scanner database has not been introduced in the paper. More information is needed.

**Deanonymize Review:**

no

**Detailed Comments:**

* What is original in the Fig 1? Please provide more detail about the datasets and scanners.


* What data augmentation steps have been used except for color variance and blur augmentation?

* Please provide more details about the training parameters that you used. Did you use the same optimizer for all networks? Did you employ Learning Rate Scheduler?

* Based on the experiments, EfficientNet B4 has the highest accuracy and higher inference time. It would be interesting to apply the network quantization techniques to this network and see if it achieves the best overall score (best trade-off between accuracy and inference speed).


**Justification Of The Rating:**

Although the statistical tests have not been provided by the authors, and as a result, this is not clear if the differences are statistically significant,  the reviewer appreciates the efforts the author put into time-consuming experiments and comparing 10 different CNNs both in terms of classification accuracy and inference time on histopathology images.

Besides, comparing different scanners has practical value for computational pathology.


**Paper Type:**

validation/application paper

**Special Issue:**

no

---

### Official Review · Reviewer_PAHw · 2021-04-30

**Confidence:** 5
**Final Rating:** 3

**Summary:**

The paper presents a comparison of robustness and inference time of multiple deep learning architectures such as EfficientNet, ResNet, DenseNet, Inception, etc., on colon cancer dataset. The paper extends the work done in [Kurticyn et al](https://www.springerprofessional.de/en/robust-slide-cartography-in-colon-cancer-histology/18909320)., demonstrating that augmentation like color, blur, etc., can help in the development of robust models performing well on multi-scanner data. Two of the significant contributions of the paper are:
* Evaluation of multiple deep learning architectures for multi-class patch classification.
* Evaluation on six different scanners dataset.


**Strengths:**

* The authors compare performance on slides obtained from six different scanners. In the recent literature, mostly the comparison has been made on slides obtained from different labs with stain variations or the same slide scanned on two different scanners. The evaluation of multiple architectures on six different scanners of the same slide presents a significant contribution.
* The paper presents an interesting comparison of the performance of multiple deep learning architectures on histopathology images. However, not much intuition or analysis of these differences has been provided in the paper.
* The mean performance of 3 runs is reported along with their standard deviation, important for comparing different architectures.


**Weaknesses:**

* No references to color augmentation literature have been provided. It will be helpful for readers if, for example, papers like [Tellez et al.](https://europepmc.org/article/med/31466046) could be cited for redirecting readers to relevant literature.
* The inference time comparison of these models is not new, as this has been done in the latest architecture paper as part of latency and FLOPS comparison ([Tan et al.](http://proceedings.mlr.press/v97/tan19a/tan19a.pdf)). The paper has extended this work and reported the mean inference time for histopathology images.


**Deanonymize Review:**

no

**Detailed Comments:**

* Interestingly, the order of model performance is not the same across different scanners. It would be interesting to study the reason behind this difference. Maybe it will help to see if the same model makes mistakes on the same or different slides across scanners.
* In the paper, the authors can highlight the gains associated with different architectures for imagenet are not directly transferable to histopathology images. One of the paper’s significant contributions is that the gains associated with different architectures seen in imagenet are not directly transferable to histopathology images.
* As mentioned above, it will be helpful if references to color augmentation literature can be added.


**Justification Of The Rating:**

The paper is clearly written and presents an interesting comparison of performance and inference time of multiple deep learning architectures on six different scanner datasets, highlighting models and augmentations that can be used for histopathology


**Paper Type:**

validation/application paper

**Special Issue:**

no

---

### Meta-Review · Program_Chairs · 2021-05-06

**Recommendation:** Accept (Poster)
**Confidence:** 5

**Metareview:**

This paper is a clear acceptance. Authors are suggested to address reviewer suggestions in final version.

---

### Decision · Program_Chairs · 2021-05-11

Accept (Poster)